# Serial Dual Mediating Effects of Parenting Stress on Life Satisfaction among Parents of School-Aged Children with Chronic Conditions

**DOI:** 10.3390/healthcare12040461

**Published:** 2024-02-11

**Authors:** Jeong-Won Han, Boeun Yang, Hanna Lee

**Affiliations:** 1College of Nursing Science, Kyung Hee University, 26, Kyunghee-daero, Dongdaemun-gu, Seoul 02447, Republic of Korea; hjw0721@naver.com (J.-W.H.); 90471361@khu.ac.kr (B.Y.); 2Department of Nursing, Gangneung-Wonju National University, 150, Namwon-ro, Heungeop-myeon, Wonju-si 26403, Republic of Korea

**Keywords:** child, chronic disease, life satisfaction, parent, self-esteem, stress

## Abstract

This study examines the serial dual mediating effects of marital conflict and self-esteem on the relationship between parenting stress and life satisfaction in parents of school-aged children with chronic conditions. This study aims to present foundational data for developing nursing interventions for parents caring for children with chronic illnesses. Of the 2150 parents who participated in the 13th Panel Study on Korean Children (PSKC), 271 raising a child with a chronic illness were enrolled in the study. The serial dual mediating effect was analyzed using PROCESS macro Model 6. The serial dual mediating effects of parenting stress, marital conflict, and self-esteem on parents’ life satisfaction were analyzed. Marital conflict and self-esteem had significant serial multiple mediating effects on the relationship between stress and life satisfaction in fathers (B = −0.11, bootstrap 95% CI = −0.16–−0.06) and mothers (B = −0.06, bootstrap 95% CI = −0.09–−0.03). Our results suggest that marital conflict increases with increasing parenting stress and that increased marital conflict sequentially reduces self-esteem, ultimately diminishing life satisfaction in parents raising a child with a chronic condition. Thus, relevant nursing interventions and social support are essential to boost the life satisfaction of parents raising children with chronic conditions.

## 1. Introduction

Globally, healthcare utilization by children is increasing in several dimensions, particularly for chronic diseases [1]. Health problems affect not only patients but also their family members due to the stress and lifestyle changes associated with the disease [2]. Moreover, children with a disease who are unable to take care of themselves require the care of their families [3]. In particular, family members of people with chronic diseases are likely to experience stress due to fatigue from long-term care and concerns about uncertain prognoses [4]. Learning that a child has a chronic disease is shocking for family members and causes the parents emotional pain and cognitive, emotional, and behavioral stress, in addition to creating a sense of crisis and imposing complex tasks and responsibilities [1].

According to Pearlin’s Framework of Caregiver’s Burden Model [5], caregivers responsible for the care of family members with chronic illnesses experience stress in their caregiving roles due to the patient’s health status and various other factors. Pearlin’s Framework of Caregiver’s Burden Model categorizes various stress factors into background and contextual factors, primary stressors, and secondary stressors. Background and contextual factors are comprehensive influences throughout the stress process, representing the characteristics of the caregiver or care recipient, with the caregiver’s health status being a particularly significant factor. Primary stressors are conditions, experiences, and activities that pose objective difficulties or subjective perceptions of difficulty faced by individuals in providing care. Secondary stressors encompass role strain, psychological tension, and psychological states that diffuse from primary stressors. Psychological tension involves issues in the psychological dimension similar to self-concept, with self-esteem being a prominent example. Each stage influences the subsequent stage, and all stress factors ultimately impact the caregiver’s physical and psychological well-being. In addition, the chronic illness of a child can lead to shifts in familial roles and place an excessive burden on family members, influencing their interactions and potentially resulting in long-term tension [6,7]. This stress leads to conflicts among family members and reduced self-esteem, ultimately increasing the caregiver burden and impairing their quality of life (QOL) [8]. Unresolved emotional distress among caregivers can limit their ability to fulfill their parental duties, which may affect the child’s adjustment to the illness and various aspects of their QOL [9,10]. As the caregiver burden ultimately adversely affects the health of the individual with the chronic condition, nurses must understand this mechanism and consider appropriate interventions to avoid contributing to adverse effects on the health of children with chronic illnesses [11]. Therefore, it is crucial to prioritize the promotion of parents’ psychological and emotional stability and to boost their life satisfaction to effectively manage chronic conditions and promote emotional stability in school-aged children.

Stress among parents of children with chronic illnesses goes beyond the momentary and temporary worries faced while performing parental roles. It encompasses a sustained state of accumulated stress [12]. However, the parenting stress of raising a child with a chronic condition differs from that experienced by the parents of a healthy child, as the child’s chronic condition induces long-term and persistent tension. Previous studies suggest that parenting stress in mothers of children with chronic illnesses can transfer to the fathers, directly affecting their QOL [13,14] and sparking marital conflicts [15,16]. This means that early nursing interventions are needed to help manage parenting stress. Moreover, self-esteem is another predictor of life satisfaction in parents of children with chronic illnesses. Self-esteem is one’s overall evaluation, feelings about one’s worth and abilities, and the degree to which individuals perceive themselves as important and capable of success [17]. Self-esteem protects individuals from stress and curtails the adverse effects of stressful situations [18]. Positively rating oneself is linked to good QOL, so self-esteem may be a contributor to parents’ eudemonic happiness [19]. That is, parents’ self-esteem diminishes when they perceive high marital conflict and parenting stress, and as a result, they may rate their QOL as low.

As such, the stress of parents of children with chronic illnesses may not only directly impact their QOL, but also indirectly contribute to marital conflicts and self-esteem. However, most existing studies have focused on the fragmental relationships of certain factors in the QOL of healthy children or parents. Therefore, it is necessary to examine serial dual mediating relationships involving all four factors. In particular, previous studies on the relationship between parental caregiving stress and QOL in children with chronic illnesses predominantly examined direct effects and simple mediating effects between variables. However, these approaches alone cannot adequately explain the complexities of social phenomena. Mediators are utilized to understand the underlying causal structure between independent and dependent variables, and serial multiple mediation involves the independent variable serially altering two or more mediators, resulting in a change in the dependent variable. Thus, it is used to investigate the effects of the mediator by examining the direct and indirect effects [20]. As noted, exploring the relationship between stress and life satisfaction in parents of children with chronic illnesses and establishing evidence to highlight the importance of psychological support for the family are of paramount importance. 

Therefore, this study aims to examine the serial dual mediating effects of marital conflict and self-esteem on the relationship between parenting stress and life satisfaction in parents of children with chronic illnesses and to present foundational data for developing nursing interventions for these parents.

## 2. Materials and Methods 

### 2.1. Study Design

This study conducted a secondary data analysis to examine the serial dual mediating effects of marital conflict and self-esteem on the relationship between parenting stress and life satisfaction in parents of school-aged children with chronic illnesses using the Panel Study of Korean Children (PSKC) conducted by the Korea Institute of Child Care and Education (KICCE) (Figure 1).

### 2.2. Study Participants

Of the 2150 parents who participated in the PSKC, 271 parent pairs, each with a child with a chronic condition, were enrolled in the study. The criteria for children with chronic illnesses were that they be school-aged children with (or with a history of) their illness for at least three months (including accidental), currently undergoing continuous treatment for the illness, and who must visit a hospital at least once a year for the illness [21]. The mean age was 45.46 years for fathers and 42.81 years for mothers. The most common education level was a bachelor’s degree (*n* = 97, 35.8%) for fathers and mothers (*n* = 114, 42.1%). The most common occupation for fathers was as a manager or in a white-collar job (*n* = 139, 51.3%). For mothers, it was as a homemaker, student, or unemployed person (*n* = 117, 43.2%), or as a manager or in a white-collar job (*n* = 111, 40.9%). The child’s gender was male (*n* = 149, 55.0%) or female (*n* = 122, 45.0%). The mean duration of marriage was 189.58 months, and the most common chronic condition was allergic disease (*n* = 206, 76.0%) (Table 1). 

### 2.3. Instruments

This study used the instruments used in the PSKC, as shown in the following subsections.

#### 2.3.1. Parenting Stress

For parental stress, the Korean version of the ‘Parenting Stress Scale’ developed by Kang [22] based on Abidin’s [23] Parenting Stress Index (PSI) was used. Among these, the children’s panel used a questionnaire that extracted only the ‘burden and distress about parental role’ factors that were judged to be closely related to parenting stress. A questionnaire containing only excerpts was used by the children’s panel. This 24-item tool consists of 17 items for stress incurred by performing the role of a parent of a school-aged child and seven items for parenting stress. It consisted of items such as “I am worried about my child’s ability to adapt to school life” and “I am stressed about my child’s academic performance”. Each item is rated on a 5-point Likert scale ranging from 1 “Strongly disagree” to 5 “Strongly agree”. A higher total score indicates greater parenting stress. The reliability (Cronbach’s α) of the instrument was 0.94 for fathers and 0.94 for mothers in this study.

#### 2.3.2. Marital Conflict

Marital conflict was measured using items restructured by the KICCE researchers with reference to Markman, Stanley, and Blumberg [24] and Chung [25]. It consists of eight items, such as “I feel like my husband doesn’t respect me” and “I seriously think about what it would be like to date or marry someone else”. There are eight items for marital conflict, and each is rated on a 5-point Likert scale ranging from 1 “Strongly disagree” to 5 “Strongly agree”. A higher total score indicates greater marital conflict. The reliability (Cronbach’s α) of the instrument was 0.92 for fathers and 0.93 for mothers in this study.

#### 2.3.3. Self-Esteem

Self-esteem was measured using the Rogenberg Self-Esteem Scale [17], which was modified and adapted by Lee [26] and further modified by the KICCE research team. There are 10 items for measuring self-esteem, and each item is rated on a 5-point Likert scale ranging from 1 “Strongly disagree” to 5 “Strongly agree”. A higher total score indicates more positive self-esteem. The reliability (Cronbach’s α) of the instrument was 0.89 for fathers and 0.91 for mothers in this study.

#### 2.3.4. Life Satisfaction

For life satisfaction, the participants’ overall satisfaction with life was measured using one question: “How satisfied are you with your current life?” The item was rated on a 5-point Likert scale ranging from 1 “Strongly disagree” to 5 “Strongly agree”. A higher total score indicates greater life satisfaction. The inclusion of a single-item measure of life satisfaction in this study is justified, being supported by both expert considerations and previous studies [27,28] that demonstrate the effectiveness of such an approach. In particular, the Korean Children’s Panel employed in our study constitutes national-level panel data, utilizing a stratified multistage sampling method for sample selection. The selection and validation of the survey items were meticulously curated through the oversight and research outcomes of expert members from the Policy Research Institute, ensuring the reliability of our measures.

#### 2.3.5. Income Level

Income level is the average monthly household income and is the sum of all types of income (e.g., work, finances, business, rental property, transfer, and other income) of all family members. Income was calculated by subtracting all relevant deductions, including income tax, local inhabitants’ tax, property tax, interest income tax, national pension, and health insurance premiums.

### 2.4. Data Collection

The 13th PSKC was used in this study. The PSKC data are available for public use. We downloaded the de-identified data (http://panel.kicce.re.kr, accessed on 1 July 2023.) after submitting a request for use to the KICCE, specifying the authors’ affiliations and the purpose of data use. This study was exempted from review by the Gangneung-Wonju National University Institutional Review Board (IRB) due to the secondary analysis of existing data and the absence of identifiable information (GWNUIRB-R2023-34).

### 2.5. Data Analysis

The collected data were analyzed using SPSS 25.0 (IBM Corp., Armonk, NY, USA). The participants’ general characteristics and study variables were analyzed with descriptive statistics. The reliability of the instruments was evaluated using the internal consistency coefficient, and the relationships among the variables were analyzed using Pearson correlation coefficients. The effects among the variables were analyzed using PROCESS macro v3.4 [20]. The PROCESS macro runs based on bootstrapping, which is known to be useful for evaluating mediations because the method does not assume normal distribution and t-distribution [29]. We used PROCESS macro Model 6 to test our serial dual mediation hypothesis. The statistical significance of the computed indirect effects was tested by bootstrapping with 10,000 samples and a 95% bias-corrected confidence interval (CI) [29]. This method involves iterative bootstrapping with the same sample size to calculate a 95% confidence interval for the mediating (indirect) effect. If this confidence interval does not include zero, the mediating (indirect) effect is considered statistically significant. The bootstrapping method is characterized by high statistical power, as it does not assume normality constraints [20].

## 3. Results

### 3.1. Correlations among Variables

The correlation between the variables is shown in Table 2.

### 3.2. Serial Dual Mediating Effect of Parenting Stress on Life Satisfaction

Fathers’ parenting stress had an effect on marital conflict (B = 0.48, *p* < 0.001) and self-esteem (B = −0.29, *p* < 0.001), and marital conflict had an effect on self-esteem (B = −0.32, *p* < 0.001) and life satisfaction (B = −0.19, *p* < 0.001). Self-esteem had an effect on life satisfaction (B = −0.19, *p* < 0.001). The serial multiple mediating effects of parenting stress, marital conflict, and self-esteem on fathers’ life satisfaction were analyzed. Marital conflict and self-esteem had a significant serial multiple mediating effect on the relationship between fathers’ stress and life satisfaction (B = −0.11, bootstrap 95% CI = −0.16–−0.06). In other words, marital conflict increases alongside fathers’ increasing parenting stress, and increased marital conflict sequentially reduces self-esteem, ultimately lowering life satisfaction (Table 3). The results of checking for multicollinearity indicated that the tolerance values ranged from 0.61 to 0.87, all of which were below 1.0, suggesting no issues with multicollinearity. Additionally, the Variance Inflation Factor (VIF) ranged from 1.15 to 1.64, not exceeding the threshold of 10. Thus, it was confirmed that there were no problems with multicollinearity. Furthermore, in the autocorrelation verification, the Durbin–Watson statistic was 2.04, close to 2, indicating the absence of autocorrelation among independent variables.

Mothers’ parenting stress affects marital conflict (B = 0.35, *p* < 0.001) and self-esteem (B = −0.32, *p* < 0.001), and marital conflict affects self-esteem (B = −0.33, *p* < 0.001) and life satisfaction (B = −0.26, *p* < 0.001). Finally, self-esteem affects life satisfaction (B = 0.39, *p* < 0.001). The serial multiple mediating effects of parenting stress, marital conflict, and self-esteem on mothers’ life satisfaction were analyzed. Marital conflict and self-esteem had a significant serial multiple mediating effect on the relationship between mothers’ stress and life satisfaction (B = −0.06, bootstrap 95% CI = −0.09–−0.03). In other words, marital conflict increases with mothers’ increasing parenting stress, and increased marital conflict sequentially reduces self-esteem, ultimately lowering life satisfaction (Table 4).

## 4. Discussion

First, parenting stress influenced marital conflict, which in turn influenced the life satisfaction of both parents. This is similar to the findings of a study conducted in China, where mothers’ parenting stress was negatively correlated with marital satisfaction, and marital stress mediated the relationship between parenting stress and marital satisfaction [15]. A study of 161 parents rearing a seven-year-old child with atopic dermatitis also reported that parents’ parenting stress affects marital conflict [16]. Another reported that the marital satisfaction of parents with a child exhibiting behavioral problems partially mediates the relationship between parenting stress and life satisfaction [30]. This suggests that the influence of parenting stress on QOL may vary depending on the mediating variable, marital conflict. The greatest source of parenting stress in parents of children with chronic illnesses is the difficulty of feeling the reward of parenting despite being primary caregivers, spending the most time with the child, and the feeling of guilt that they are not doing much to help their child’s development [7]. Children with chronic conditions and their families often face a lack of social understanding, empathy, and support. As shown here, married couples raising children with chronic illnesses often face threats to their marital relationships due to feelings of anxiety, guilt, and the stress related to social and cultural factors associated with caregiving. They are also reported to show higher rates of marital dissatisfaction and depressive symptoms than parents of typically developing children [6], highlighting the need for interventions that address parental caregiving stress and marital conflict in married couples raising a child with a chronic condition. Effective communication between spouses plays a vital role in overcoming the challenges faced during crisis situations [7]. Thus, it is crucial to develop communication programs that help couples navigate marital conflicts more effectively. Emotional support is crucial for parents and families of children with chronic conditions. Fostering understanding and minimizing negative impacts while promoting positive experiences is necessary. Forming a sense of community and open communication is vital to prevent social isolation and help families handle challenging situations. Relationships in local communities and among family members serve as powerful social networks that enhance overall QOL. Furthermore, a study on mothers of elementary school children with allergies emphasized the importance of combining pharmacological treatment with psychosocial counseling and therapy for both the child and the caregiver [31]. Therefore, developing educational programs, counseling, and marital communication interventions for parents of children with chronic illnesses would be a promising approach to improving their overall QOL.

Second, parenting stress influenced self-esteem, which affected the life satisfaction of both parents. These results are similar to the findings of a Korean study conducted on 656 children diagnosed with allergic rhinitis and their parents, which found that fathers’ parenting stress influenced mothers’ self-esteem [32]. Furthermore, these results support the findings that positively evaluating oneself is linked to one’s QOL [33]. Since self-esteem is strongly correlated with QOL [18,34], it can be inferred that QOL is impaired by decreased self-esteem when marital conflict and parenting stress are heightened. Therefore, individuals with high self-esteem can prevent a negative path to quality of life in situations with high parenting stress. Whereas individuals with high self-esteem tend to have positive self-perceptions and self-worth, and thus actively engage in social roles, parents with low self-esteem tend to evaluate themselves negatively and exhibit passive attitudes [34]. Therefore, although environmental conditions or situations cannot be modified, parents of children with chronic illnesses may attain better QOL by developing higher self-esteem.

Third, serial multiple mediation describes the impact of the independent variable on the dependent variable by sequentially altering two or more mediators. The key objective of this analysis is to elucidate the effects of the mediators by examining both direct and indirect effects [20]. A mediating effect implies that the change in the dependent variable is more substantial when the change in the independent variable is coupled with changes in the mediators. In our study, we discovered that marital conflict and self-esteem have a serial mediating effect on the relationship between parental stress and life satisfaction in both mothers and fathers. This signifies that heightened parenting stress leads to increased marital conflict, which subsequently diminishes self-esteem and ultimately decreases the life satisfaction of mothers and fathers. These serial mediating effects highlight that the magnitude of change in the life satisfaction of parents of a school-aged child with a chronic condition is more pronounced when parents’ marital conflict and self-esteem are moderated than simply addressing their parenting stress alone. Thus, interventions that effectively reduce parenting stress, mitigate marital conflict, and foster improved self-esteem are required to improve life satisfaction and overall quality of life in parents of children with chronic illnesses. This is because while it is difficult to change the physical home environment, modifying functional aspects, such as communication and parental support, can lead to decreased marital conflict and increased self-esteem, ultimately leading to heightened life satisfaction. To this end, it is important to provide parental education and counseling support for parents of children with chronic illnesses at the school, community, and national levels. Moreover, education programs pertaining to the development of children with chronic conditions, marital communication skills, and the improvement of self-esteem should be developed and promoted in healthy family support centers available nationwide. 

This study had several limitations. Because we used secondary data, many variables could not be examined. In the future, additional data can be collected to supplement insufficient variables, or various research methodologies can be applied to address the problem from various aspects. In addition, our study utilized a brief single-item measurement tool to assess QOL, which limited the depth of our data. This simplicity may ignore nuanced aspects of QOL, potentially compromising accuracy. Caution is advised in interpreting and generalizing our findings due to the inherent constraints of our singular measurement approach. Future research may consider employing more comprehensive multi-item scales for a nuanced understanding of QOL.

## 5. Conclusions

This study evaluated the serial multiple mediating effects of marital conflict and self-esteem on the relationship between parenting stress and life satisfaction in parents of children with chronic conditions. The results confirmed that marital conflict and self-esteem serially mediate the relationship between stress and life satisfaction in mothers and fathers. In other words, marital conflict increases with increasing parenting stress, which serially reduces self-esteem and ultimately diminishes life satisfaction. One key implication of this study is that it provided an empirical basis for nursing research or interventions to promote life satisfaction in parents raising children with chronic conditions. Based on our findings, we offer three recommendations for future studies. First, studies should further examine various psychological factors that predict the QOL of parents of children with chronic conditions. Second, subsequent studies should establish a longitudinal study model that considers temporal differences to examine the relationships among the variables and their changes over time. Third, interventions to mitigate parenting stress, reduce marital conflict, and improve self-esteem should be developed and implemented to enhance the life satisfaction of parents of school-aged children with chronic conditions.

## Figures and Tables

**Figure 1 healthcare-12-00461-f001:**
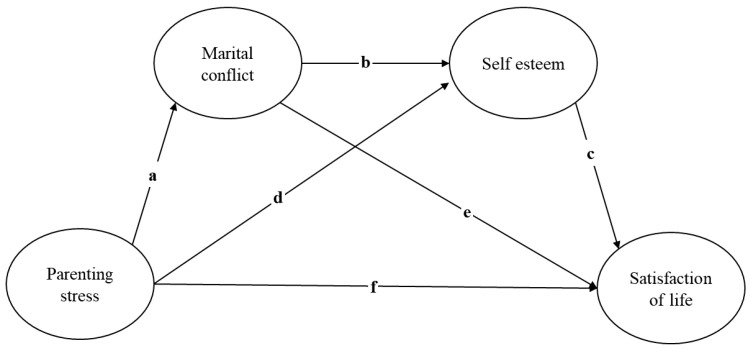
Framework.

**Table 1 healthcare-12-00461-t001:** General characteristics (*n* = 271 dyad).

Variables	Categories	Paternal	Maternal
		n (%) or M ± SD	n (%) or M ± SD
Age (yr)	≤40	23 (8.7)	66 (29.4)
	41–45	114 (34.9)	152 (51.3)
	≥46	134 (56.4)	53 (19.3)
		45.46 ± 3.60	42.81 ± 3.31
Education level	High school	66 (24.4)	60 (22.1)
	College	68 (25.1)	76 (28.0)
	Bachelor’s degree	97 (35.8)	114 (42.1)
	Master’s degree or higher	40 (14.7)	21 (7.8)
Occupation	Manager or white-collar job	139 (51.3)	111 (40.9)
	Service sector or sales person	29 (10.7)	29 (10.7)
	Agriculture or forestry and fishery	2 (0.7)	0 (0.0)
	Engineer or machine fabricator	64 (23.6)	11 (4.1)
	Simple labor	5 (1.9)	3 (1.1)
	Unemployed	32 (11.8)	117 (43.2)
Gender of child	Male	149 (55.0)	
	Female	122 (45.0)	
Marriage period		189.58 ± 32.12	
Chronic disease type of child	Congenital heart disease	8 (3.0)	
	Epilepsy	2 (0.7)	
	Diabetes	2 (0.7)	
	Chronic sinusitis	15 (5.6)	
	Chronic otitis media	4 (1.4)	
	Asthma	14 (5.1)	
	Allergic disease	206 (76.0)	
	Atopic dermatitis	20 (7.5)	

M: mean, SD: standard deviation. Unemployed individuals include students and homemakers.

**Table 2 healthcare-12-00461-t002:** Correlations among variables.

Variables (Paternal)	M ± SD	X1	X2	X3	X4	X5
Family income (X1)	537.36 ± 198.90	1				
Parenting stress (X2)	2.31 ± 0.65	−0.09 *	1			
Marital conflict (X3)	2.07 ± 0.76	−0.02 *	0.39 *	1		
Self-esteem (X4)	3.80 ± 0.57	0.15 *	−0.45 *	−0.54 *	1	
Satisfaction with life (X5)	3.50 ± 0.78	0.32 *	−0.27 *	−0.47 *	0.61 *	1
Variables (Maternal)	M ± SD	X1	X2	X3	X4	X5
Family income (X1)	537.36 ± 198.90	1				
Parenting stress (X2)	2.50 ± 0.65	−0.14 *	1			
Marital conflict (X3)	2.13 ± 0.81	−0.03 *	0.32 *	1		
Self-esteem (X4)	3.74 ± 0.62	0.21 *	−0.43 *	−0.45 *	1	
Satisfaction with life (X5)	3.38 ± 0.77	0.27 *	−0.31 *	−0.45 *	0.54 *	1

M: mean, SD: standard deviation. * *p* < 0.01.

**Table 3 healthcare-12-00461-t003:** Sequential mediating effects of parenting stress on satisfaction with life (paternal).

Path	B	SE	t	*p*	LLUI	ULCI	R^2^	F	*p*
Parenting stress → Marital conflict	0.48	0.07	6.88	<0.001	0.34	0.62	0.17	23.80	<0.001
Parenting stress → Self-esteem	−0.29	0.05	−6.07	<0.001	−0.38	−0.19	0.42	55.94	<0.001
Marital conflict → Self-esteem	−0.32	0.05	−7.85	<0.001	−0.40	−0.24			
Parenting stress → Satisfaction with life	0.05	0.07	0.64	0.52	−0.09	0.19	0.40	38.48	<.001
Marital conflict → Satisfaction with life	−0.19	0.06	−3.05	<0.001	−0.32	−0.07			
Self-esteem → Satisfaction with life	0.70	0.09	7.58	<0.001	0.52	0.88			
Parenting stress → Satisfaction with life (Total effect)	−0.35	0.07	−4.82	<0.001	−0.50	-0.21	-		
Indirect effect path	Effect		Boot SE		Boot LLCI			Boot ULCI	
Parenting stress → Marital conflict → Satisfaction with life	−0.09		0.04		−0.18			−0.03	
Parenting stress → Self-esteem → Satisfaction with life	−0.20		0.05		−0.29			0.11	
Parenting stress → Marital conflict → Self-esteem → Satisfaction with life	−0.11		0.03		−0.16			−0.06	

SE: standard error, LLCI: lower limit of the 95% confidence interval, ULCI: upper limit of the 95% confidence interval.

**Table 4 healthcare-12-00461-t004:** Sequential mediating effects of parenting stress on satisfaction with life (maternal).

Path	B	SE	t	*p*	LLUI	ULCI	R^2^	F	*p*
Parenting stress → Marital conflict	0.35	0.07	5.02	<0.001	0.29	0.58	0.12	17.36	<0.001
Parenting stress → Self-esteem	−0.32	0.05	−5.57	<0.001	−0.41	−0.20	0.29	33.43	<0.001
Marital conflict → Self-esteem	−0.33	0.04	−5.75	<0.001	−0.34	−0.17			
Parenting stress → Satisfaction with life	−0.05	0.07	−0.92	0.36	−0.20	0.07	0.36	34.39	<0.001
Marital conflict → Satisfaction with life	−0.26	0.06	−4.51	<0.001	−0.37	−0.14			
Self-esteem → Satisfaction with life	0.39	0.08	6.47	<0.001	0.35	0.66			
Parenting stress → Satisfaction with life (total effect)	−0.39	0.07	−5.30	<0.001	−0.53	−0.24	-		
Indirect effect path	Effect		Boot SE		Boot LLCI			Boot ULCI	
Parenting stress → Marital conflict → Satisfaction with life	−0.11		0.04		−0.20			−0.05	
Parenting stress → Self-esteem → Satisfaction with life	−0.15		0.04		−0.22			−0.08	
Parenting stress → Marital conflict → Self-esteem → Satisfaction with life	−0.06		0.02		−0.09			−0.03	

SE: standard error, LLCI: lower limit of the 95% confidence interval, ULCI: upper limit of the 95% confidence interval.

## Data Availability

The datasets used and/or analyzed during the current study are available from the corresponding author upon reasonable request.

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
