# Peer review of "Serial Dual Mediating Effects of Parenting Stress on Life Satisfaction among Parents of School-Aged Children with Chronic Conditions"

_healthcare, 2024, doi:10.3390/healthcare12040461_

Round 1
Reviewer 1 Report
Comments and Suggestions for Authors
1. The literature review need revise substantially. It does not provide adequate information about the relationships of different variables in your study.
2. The discussion of theoretical framework is very weak. It should be revised substantially.
3. The research question and research objectives are not clearly described in the manuscript.
4. More description on the scale you are using is desirable.
5. In data analysis, please report any collinearity among the variables in the model.
6. Please describe in detail how this piece of study is contributing to the extand literature and advancement in the theory.
Comments on the Quality of English LanguageI suggest the manuscript to be edited by native English editor before publication.
Author Response
Dear Editor:
Thank you for reviewing the paper “ Serial dual mediating effects of parenting stress on life satisfaction among parents of school-aged children with a chronic condition Parenting stress” submitted for review for publication in the Healthcare. I would also like to thank the anonymous reviewers for their comprehensive and helpful comments. The manuscript has been revised accordingly. We hope that we have sufficiently addressed the concerns raised by the reviewers and we have attached a certificate of English editing for the revised paper. We look forward to hearing from you. Please let us know if you have any further questions. Thank you for your kind attention.
Sincerely,
Authors

Reviewer 2 Report
Comments and Suggestions for Authors
Thank you for the opportunity to read this very interesting paper. I have a few suggestions for revisions although I do not think they will be at all difficult to address:
1. on p. 2 you make reference to epidemic - was this meant to tell us the research was done during the COVID -19 period? It is unclear and the connection is not explained nor does it appear as a feature elsewhere.
2. In terms of the instruments used, please confirm that all measures have been validated on the Korean population and thus are valid for the purpose.
3. In terms of the social place of disabled children, is there a cultural factor in how such children are perceived? This can be a source of further marital stress.
4. In the discussion, there is reference to a variety of formalized supports / programs. I wonder if there should be some consideration given to informal supports such as family members

Author Response

(The authors gave the same response as above.)

Reviewer 3 Report
Comments and Suggestions for Authors
I have several observations about this paper:
1. The introduction is densely written and needs to provide a rationale for the present paper-what unique knowledge is it providing? What cited literature suggests the need for the present study? What studies are providing "fragmental relationships of certain factors in the QOL of healthy children or parents?" What factors? The intro is infused with recommendations that belon in the discussion. Thus, the rationale needs strengthening and needs to be more concise.
2. More information needs to be provided regarding the PROCESS analysis. What does it accomplish? Exactly how was it utilized?
3. A significant limitation not recognized by the authors is the single item index of life satisfaction. Its use is indefensible-there are no psychometrics or validity findings for this item. This is an artifact of the use of secondary data-that it is secondary data does not weaken the measurement problem here.
4. The sample characteristics belong in the methods section, not results.
5. Given that the correlations are tabularized, there is no need to present them in the text.
6. Like the introduction, the discussion is densely written. Stick to what the findings mean.
7. I see virtually no acknowledgement of any limitation (1 sentence) in the limitations section and the conclusion section is not conclusive-it needs to be relabeled as recommendations for research and practice-what is in the discussion that overlaps with this should be deleted.
Author Response

(The authors gave the same response as above.)

Round 2
Reviewer 1 Report
Comments and Suggestions for Authors
Thanks for addressing all my concerns. I am satisfied with the revised version.
Author Response
Thank you for reviewing my paper. I appreciate your time and effort.
Reviewer 3 Report
Comments and Suggestions for Authors
Despite the changes by the authors which have improved the paper, the use of a single item index of life satisfaction is still not defensible. Acknowledging this (which is really not well spelled out) does not lessen it as a central weakness of the paper. My recommendation is that this be omitted and focus on the joint effects of marital conflict and parenting stress on self esteem.
Comments on the Quality of English LanguageDespite the changes by the authors which have improved the paper, the use of a single item index of life satisfaction is still not defensible. Acknowledging this (which is really not well spelled out) does not lessen it as a central weakness of the paper. My recommendation is that this be omitted and focus on the joint effects of marital conflict and parenting stress on self esteem.
